# SERDES Link Training with Edge Inference: Neural-Network Driven Discrete Optimization to Maximize Link Efficiency

## Abstract

Meeting the growing data demands of modern AI applications requires efficient, high-speed communication links. We propose an edge inference framework that dynamically optimizes non-uniform quantization levels in programmable ADC receivers. While integer linear programming (ILP) offers high-quality solutions, its significant computational cost (120 seconds per instance on high-performance CPUs) and hardware requirements make it unsuitable for on-chip use. On-chip solutions are essential for fast, periodic adjustments to track time-varying effects such as temperature drift and ensure reliable communication. To address this, we train a convolutional neural network (CNN) using ILP-generated labels, achieving a 24,000x speedup with inference on a RISC-V microcontroller. The CNN leverages a custom loss function tied to system-level metrics, reducing area metric errors from 29% to less than 2%. Unlike prior works embedding neural networks in the signal path, our framework adapts periodically to channel variations without disrupting communication. This enables improved error rates, energy efficiency, and a scalable pathway for on-chip edge intelligence in next-generation systems.

## 1 Introduction

As AI models continue to expand at an unprecedented rate, with modern architectures containing billions or even trillions of parameters (Fig. 1(a)), the demands on the underlying **data communication and high-speed links** have also grown commensurately.

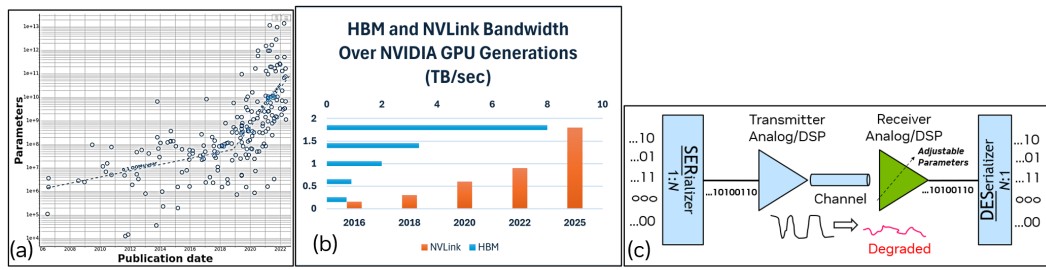

Figure 1: (a) The exponential growth of AI model parameters over time, driving increasing demand for high-speed data communication. (b) Bandwidth growth for NVLink and HBM SERDES across NVIDIA GPU generations, showing how communication infrastructure is scaling to meet these demands (c) High-level SERDES link diagram showing how signal degradation occurs over the channel, emphasizing the role of the receiver in adapting its parameters to ensure accurate signal detection

Figure 1(b) highlights how high-speeds links such as **NVLink** and **HBM** bandwidth have scaled over time to meet the increasing data transfer requirements of AI systems. However, as data rates increase, maintaining error-free communication becomes more challenging. Both NVLink and HBM, along with other high-speed interfaces, rely on **SERDES** (Serializer/Deserializer) technology to convert parallel data into serial form for transmission over a channel and then back into parallel data

at the receiver (Fig. 1(c)). As signals pass through the channel, they are subject to attenuation and noise, leading to degraded signal quality. Furthermore, time-varying impairments such as temperature drift further impact the signal integrity. All together, these impairments create a heavy burden for the receiver to accurately recover the transmitted data.

To mitigate these issues, the receiver needs to dynamically adjust key parameters to effectively decode the degraded signals. To address these challenges, we propose a **machine learning-based framework** that leverages a Convolutional Neural Network (CNN) to optimize the receiver's parameters periodically. Our approach ensures that the receiver can dynamically adapt to varying signal conditions to maximize link performance.

Figure 2 presents a high-level overview of our system architecture and methodology. The receiver architecture features an analog-to-digital converter (ADC) with $k$ non-uniform levels (b). A pattern buffer stores previous received data, and a look-up table (LUT) assigns one of the $k$ levels to each pattern case (c). With $m$ feedback taps in the buffer, the LUT contains $2^m$ entries. With the use of pilot training sequences, known data is transmitted, and errors are recorded in 2D eye matrices indexed by pattern cases (d). The goal is to determine the optimal values for both the $k$ levels and LUT entries in an online fashion. The sections that follow break down each component and step of our design and methodology in greater detail.

- **Background and Related Work**: Section 2 provides a brief overview of receiver design and conventional optimization techniques. We then discuss machine-learning approaches for high-speed links and edge inference applications.

- **CNN Model and Problem Formulation**: In Section 4, we discuss the CNN architecture (Fig. 2(g)) used to predict optimal ADC slice levels and LUT entries for the receiver (Fig. 2(h)). In Section 3, we formulate the underlying discrete optimization problem, where the labels for the CNN are generated by an ILP solver (Fig. 2(e)).

- **Training Pipeline**: In Section 4, we discuss our CNN training details including a custom loss function which significantly outperforms standard metrics like cross-entropy and MSE. In Section 5 we showcase our training results.

- **Edge Inference with Microcontroller**: Section 6 discusses our CNN implementation on a Risc-V microcontroller including deisgn considerations such as area and latency.

- **Performance Evaluation**: Finally, in Section 7, we show the results of our approach using measurement data on a few systems, and discuss the potential gains over conventional schemes.

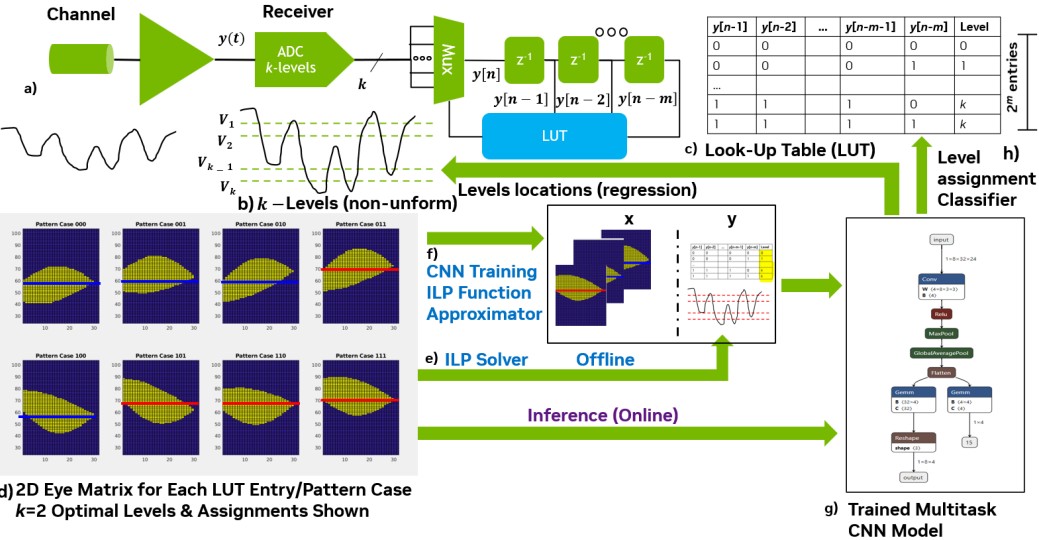

Figure 2: High-Level Summary of Receiver Design and Link Training Framework

## 2 BACKGROUND AND RELATED WORK

In high-speed communication links, signals are affected by inter-symbol interference (ISI), crosstalk, and random noise. We define this in Eqn. 3 where $x_j[n-m]$ are the transmitted symbols, $J$ is the number of lanes, $M$ is the number of prior symbols, $T$ is the symbol period, and $\eta(t)$ is random noise. Figure 3(a) shows the characterization of channel's ISI and crosstalk pulse response ($p(t)$). The noise free pulse response would be a $\delta$-function, but clearly we see signal energy spread in time and in space from adjacent signals (crosstalk).

$$y_\nu(nT+t) = \sum_{j=1}^{J} \sum_{m=0}^{M} x_j[n-m] \cdot p_{j,\nu}(t+mT) + \eta(t), \tag{1}$$

An example of this continuous time representation is shown in Fig. 3(b). The quality of the received signal can be visualized using an eye diagram which folds the signal at each clock cycle boundary (Fig. 3(c)). The "eye" opening represents the margin for error-free detection. A larger eye-opening indicates a clearer distinction between transmitted bits, while a smaller eye indicates more signal degradation due to ISI, crosstalk, and noise. For state-of-the-art (SOTA) high-speed links, the eye is often "closed," necessitating advanced equalization and digital signal processing (DSP) techniques to "open" the eye. Ultimately, the receiver performs analog-to-digital conversion (ADC), converting the analog signal into a stream of binary data.

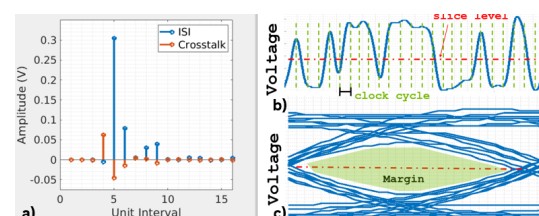

Figure 3: Link Fundamentals (a) Pulse response $p(t)$ (b) continuous-time received signal $y(t)$ (c) eye diagram visualization

### 2.1 TARGET LINKS AND ADC LEVELS

In modern long-reach SERDES designs, dedicated ADC blocks typically employ fixed, uniform quantization levels, followed by extensive DSP blocks such as feedforward and decision feedback equalizers (FFE/DFE) and maximum likelihood sequence detectors (MLSD) to recover the signal. In contrast, our work focuses on shorter-reach interfaces, such as memory links (LPDDR, GDDR, DDR) and chip-to-chip interconnects over PCB or module substrates. These interfaces often utilize simpler receivers with lower effective number of bits (ENOB $\leq \log_2(k)$) ADCs, or in some cases, no explicit ADC circuits at all.

While evaluating higher ENOB ADCs ($ENOB > 3$) is a valuable research direction, our focus on shorter-reach links is motivated by their distinct tradeoffs and their prevalence in modern computing platforms. By targeting systems with lower ENOB ($ENOB < 3$, $k < 8$), we aim to reduce power consumption and hardware complexity. This is achieved by optimizing non-uniform ADC levels and employing LUT-based signal detection to improve receiver efficiency. Further details on receiver design trade-offs and architectural differences are provided in Section A.1.

### 2.2 HIGH-SPEED LINK HARDWARE PARAMETER DERIVATION

Table 1 summarizes established approaches for determining high-speed link parameters. The simplest method is "characterization," which measures several parts to determine a best-known value (BKV) for all shipped parts. While this approach is low in complexity, it cannot track static or dynamic variations since a fixed BKV is used. To address variations, most high-speed links rely on

Table 1: Link Parameter Derivation Approaches

| Approach | Variation | | Efficiency | Complexity |
| --- | --- | --- | --- | --- |
| | Static | Dynamic | Impact | |
| Characterization | No | No | None | Low |
| Training | Yes | Yes | Yes | Moderate |
| Adaptation | Yes | Yes | None | High |

either link training or adaptation loops. Link training interrupts the link to send known data and optimize parameter settings Proakis (2007), whereas adaptation-based methods use redundant hardware, such as sampling circuits, paired with efficient algorithms like Sign-Sign Least Mean Squares (SS-LMS) Sayed (2003), to refine parameters continuously. However, these circuits increase SERDES area and power. Our approach uses neural techniques to enhance link training with minimal microcontroller hardware overhead. Before detailing our problem formulation, we review related machine learning and AI applications in high-speed communication systems.

## 2.3 Related Work: Applications of ML and AI to High-Speed Receiver Design

Machine learning has been applied to high-speed communication for tasks like signal detection and transceiver optimization. Unlike theoretical studies on end-to-end system optimization (e.g., (O'Shea & Hoydis, 2017; Zappone et al., 2019; He et al., 2019)), our work focuses on practical link parameter derivation in hardware. Related works, summarized in Table 2 (e.g., Samiee et al. (2020), Li et al. (2022), Kim (2023)), primarily integrate neural techniques within the signal path.

For example, *Deep ADC* and *NeuralEQ* use neural networks for ADC quantization and symbol detection, respectively. In contrast, our approach **decouples the neural network from the signal path**, leveraging it to optimize receiver parameters like ADC levels and LUT mappings for indirect performance gains. Similarly, *NeuADC* uses RRAM conductance tuning within ADCs, whereas we rely on software-based optimization.

Unlike real-time continuous methods, our framework performs **periodic updates**, efficiently adapting to slow time-varying effects (e.g., temperature drift) while minimizing power consumption. Moreover, our approach uses pilot training sequences, ensuring robust parameter optimization compared to live-data reliance in other works.

Table 2: Comparison of Machine Learning Approaches for High-Speed Communication Links

|  | **This Work** | **Deep ADC (2020)** | **NeuADC (2022)** | **NeuralEQ (2023)** |
|---|---|---|---|---|
| **Target Application** | Wireline/Optical links | Wireless links | Low-speed ADCs | Wireless/Optical links |
| **Rx/ADC Clock Freq.** | $\geq$ **5**GHz | 1.024GHz | 0.3/1GHz | $\geq$ **1 0**GHz |
| **Inference Task** | ADC levels LUT Entries | ADC code | ADC quantization | Symbol detection |
| **HW Parameters Tuned** | ADC Levels LUT mapping | *None* | RRAM conductances | *None* |
| **NN Input Data** | 2D error matrices | Time-series data | 1 analog sample | Time-series data |
| **Training Labels** | ILP Solver results | Transmitted symbols | Simulated ADC levels | Transmitted symbols |
| **NN Architecture** | Multi-task CNN | Conv. + LSTM | Single hidden layer | Single hidden layer |
| **Loss Function** | Custom loss (BQM, MSE) | Missing | BER minimization | Cross-Entropy |
| **Inference Hardware** | RISC-V uController | *Unspecified* | RRAM array | *Unspecified* |
| **Inference Data** | Pilot sequences | Live data | Single sample | Live data |
| **Inference Periodicity** | Low freq ($<$1KHz) | Continuous | Continuous | Continuous |
| **Validation** | Limited | No | No | No |

We explored related work on edge inferencing as our approach targets deployment on microcontrollers. Notably, frameworks like *TensorFlow Lite for Microcontrollers* enable efficient machine learning models to run on low-power devices, further supporting the feasibility of our approach TensorFlow-Team (2019). Unlike knowledge distillation (Hinton et al., 2015), which transfers knowledge from a large teacher model to a smaller student model, our approach uses ILP to generate hardware-specific labels for a CNN, focusing on system-level optimization.

## 3 Link Performance Maximization with Discrete Optimization

As illustrated in Figure 2, parts (b) and (c) show the ADC slice levels and their locations, which serve as tunable parameters in our optimization framework. Given the multiple sources of voltage

and timing errors in high-speed links, we propose using a 2D eye area metric as it provides a robust representation of margin in both time and voltage dimensions (see Appendix). To capture this, we perform a nested sweep across time and voltage, tracking errors during a training sequence. This results in $2^m$ error counters corresponding to the various observed pattern cases $(y[n-1], ...y[n-m])$. While the continuous-time domain margin is often visualized as an eye diagram, we define our *bivariate quality metric* (BQM) as the number of points in a 2D grid of voltage and timing that achieve a bit error rate (BER) below a specified threshold $\kappa$ ($\sum_v \sum_t BER(v,t) < \kappa$). This BQM, rather than BER alone, becomes the objective in our discrete optimization approach. Fig. 2(d) illustrates the BQM concept (all yellow squares are passing locations) across the different $2^m$ pattern cases.

Consider $\mathbf{A} \in \mathbb{Z}^{2^m \times p \times n}$, a three-dimensional matrix representing the 2D error counts across the $2^m$ pattern cases. By applying a binary transformation function $T$, where each element $a_{ijk}$ of $\mathbf{A}$ is transformed such that $T(a_{ijk}) = 1$ if $a_{ijk} < \kappa$ and $T(a_{ijk}) = 0$ otherwise, we obtain the binary quality matrix $\mathbf{Q}$.

The function $\mathbf{S}$ plays a critical role in our optimization process. Mathematically, $\mathbf{S}$ can be defined as a function that selects $k$ unique values from the range $\{1, \ldots, n\}$ and assigns these levels to each of the $2^m$ pattern cases:

$$\mathbf{S} : \{1, \ldots, 2^m\} \to \{1, \ldots, n\}, \quad \text{with } \mathbf{S}(i) \subset \{1, \ldots, n\} \text{ and } |\mathbf{S}(i)| = k \quad \forall i$$

$S$ gives the slice level that should be used for the $2^m$ pattern case and ensures that each pattern case uses exactly one of the selected $k$ levels to maximize the $BQM$.

A vertical shift transformation $\mathcal{V}$, utilizing the level assignments from $S$, is applied to each 2D slice of $\mathbf{Q}$ to align all selected levels: $\mathbf{C} = \bigcap_{i=1}^{2^m} \mathcal{V}(Q_{i,:,:}, S)$, where $Q_{i,:,:}$ is the $i$-th 2D slice of $\mathbf{Q}$ post-alignment. The final optimization objective, aimed at maximizing the alignment quality across all slices, is given by summing over all x, y pixels which are error free in all slices:

$$\max_{\mathbf{S}} \sum_{x=1}^{p} \sum_{y=1}^{n} \left( \bigcap_{i=1}^{2^m} \mathcal{V}(Q_{i,:,:}, S)_{x,y} \right)$$

This expression illustrates the dual role of $S$—selecting $k$ levels and assigning a level to each slice—and directly links it to the optimization goal by computing the intersections of vertically shifted binary matrices based on the selections and assignments made by $S$.

### 3.1 Solving for $S$ Using Integer Linear Programming

To determine $\mathbf{S}$ as $m$ and $k$ increase, we utilize ILP solvers, given their robust capability to handle discrete decision variables, their ability to guarantee optimal solutions and provide efficient solutions for large-scale, high-dimensional problems Wolsey (1998); Nemhauser & Wolsey (1988); Bertsekas (2005). The pseudocode for our formulation is presented in Algorithm 1. Here, binary decision variables $X[i,l]$, $W[j,z]$, and $U[l]$ $\forall i, l, j, z$ are defined, where $X[i,l]$ indicates whether level $l$ is chosen for pattern $i$, $W[j,z]$ represents error-free locations, and $U[l]$ indicates which levels are selected. After evaluating various ILP solvers, we selected Gurobi Gurobi Optimization, LLC (2023) branch and cut solver for its superior speed.

To illustrate, consider an example with $k = 4$ and $2^m = 16$. Figure 4 visualizes the results, including annotated levels and class assignments. The red lines in Fig. 4(a) indicate the optimal slice level for $k = 1$, while the green lines represent optimal levels for $k = 4$. The red squares show the passing taps in the BQM for $k = 1$, and we observe significant enhancements in the BQM for $k = 4$ as evidenced by the green squares. Keep in mind $k$ is the number of slice levels in the reciever which is proportional to the power and design complexity, so we want to minimize this as much as possible.

## 4 Neural Networks to Predict ILP Solver Outputs $S$

As discussed in the previous section, ILP solvers are highly effective at determining the optimal $S$ function.

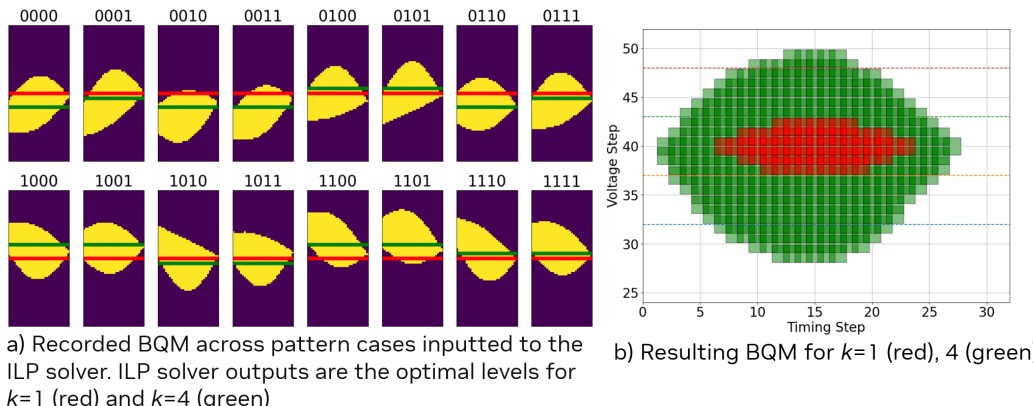

a) Recorded BQM across pattern cases inputted to the ILP solver. ILP solver outputs are the optimal levels for *k*=1 (red) and *k*=4 (green)

b) Resulting BQM for *k*=1 (red), 4 (green)

Figure 4: ILP results for $k = 4$ (a) Error counter BQM for each pattern case with $k = 1$ slice level (red) and $k = 4$ optimal slice level (green) (b) Final BQM for original $k = 1$ (red) and $k = 4$ (green)

However, implementing ILP solvers in hardware poses significant challenges, particularly in the constrained environment of high-speed SERDES links. Typically, the area allocated to SERDES controllers is minimal, and their microcontrollers handle only simple state machines and logic. This makes integrating ILP solvers impractical. To overcome this, we explore the use of neural networks to approximate ILP solver behavior, leveraging their ability to fit within small hardware footprints, as discussed in the related work section.

Given the goal of finding optimal parameters during hardware link training, we investigated supervised learning techniques to learn the ILP solver behavior. We believed this to be a solid approach given the universal function approximation properties of neural networks, ensuring that they can theoretically model any function given sufficient data and network complexity Hornik et al. (1989). With this approach, we train a neural network with the eye histogram data aggregated across phase and voltage sweeps from Section 3.1 and then use the ILP solver outputs including the optimal threshold levels and LUT entries as training labels. If successful, we can then perform our 2D BQM sweep during link training, record the error counter data, and run inference in an online fashion. Referring to Table 1, this will allow us to track part-part variation and also time varying behavior like voltage noise or temperature.

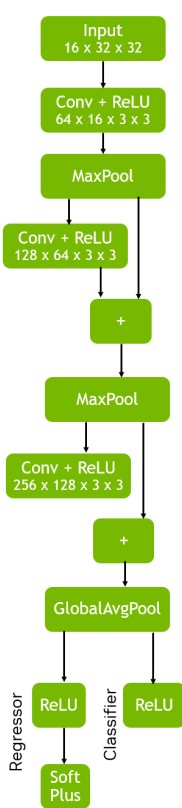

We chose to use a CNN for our application. CNNs have been very successful in image recognition tasks starting from initial work on AlexNet Krizhevsky et al. (2012) based on their ability to extract and learn robust features from complex image data LeCun et al. (1998). As a result, they are well-suited to analyzing the pass/fail regions in our 2D BQM data. This boundary detection needs to be performed across the 3rd dimension of pattern cases similar to identifying features in RGB images Goodfellow et al. (2016). While implementing convolutional layers was straightforward, determining the optimal structure for solving the problem to derive **S**—the outputs of the ILP solver—posed a greater challenge. To address this, we designed our network to handle multi-task learning, incorporating one output branch for determining the $k$ level magnitudes as a regression task, and another branch for classifying the $2^m$ pattern cases. The architecture of our multi-task network is depicted to the right in Fig. 5.

Figure 5: Multi-Task CNN architecture

Referring back to Section 3.1, we leverage the results from the ILP solver for the binary decision variables $X$ and $U$ to generate labels for our supervised CNN training. For instance, consider a scenario where $k = 4$ and $2^m = 16$. In this case, our classification labels will consist of $k = 4$ categories, represented by $0, 1, 2, 3$, while the regression targets will capture the magnitudes, which are derived from the positive integer set $\mathbb{Z}^+$.

To optimize our network, we focus on minimizing a combined loss function $L$ that incorporates both regression and classification errors, directly aligned with the outputs from our ILP solver. A conventional approach for $L$ would be to combine the losses from the regression and classifier branches where $L = L_{regression} + L_{classifier}$:

$$L = \text{MSE}(y^*_{reg} - y_{level}) + \text{BCE}(y^*_{class_{pred}}, y_{class})$$

where $y^*_{reg}$ represents the predicted regression outputs and $y^*_{class_{pred}}$ denotes the predicted probabilities for the binary classification task (or multi-class when $k > 2$). However this formulation does not capture the true metric we are after, namely the resulting BQM when applying predictions $y^*_{reg}$ and $y^*_{class_{pred}}$ to select the slice level locations and assignment to the pattern cases.

## 4.1 Custom Loss Formulations to Capture BQM

To effectively incorporate the resulting composite BQM as a loss function component, we must convert the CNN's predicted probabilities into discrete decisions and combine these with regression predictions to influence the BQM represented in a 3D tensor $x$. The transformation of probabilities into hard decisions presents a significant challenge, as it renders the loss function non-differentiable, thereby obstructing essential gradient-based optimizations. To address this, we employ the Gumbel-Softmax technique, which approximates discrete variable sampling with differentiable operations, thus maintaining the network's trainability Jang et al. (2016); Maddison et al. (2016).

Furthermore, the regression predictions, being real numbers, necessitate an affine transformation to map these continuous values effectively into our model's discrete operational framework. This is achieved using grid sampling and interpolation techniques, ensuring the preservation of differentiability. The expected shifts $\mathbf{E}$, calculated as:

$$\mathbf{E} = \sum_{j=1}^{k} \mathbf{y_{GSclass}}_j \cdot \mathbf{y_{reg}}_j$$

are applied to the BQM matrix using an affine transformation matrix $\boldsymbol{\theta}$, which adjusts each slice vertically based on the normalized expected shifts:

$$\boldsymbol{\theta} = \begin{bmatrix} 1 & 0 & 0 \\ 0 & 1 & -\mathbf{E}_{\text{norm}} \end{bmatrix}$$

This matrix alters the grid of the BQM tensor, and the subsequent processing involves an element-wise product across all $2^m$ pattern cases, synthesizing the collective effects into a scalar value representing the overall adjustment:

$$Q'' = \prod_{i=1}^{2^m} Q'_{i,:,:}, \quad BQM_{final} = \sum_{x,y} Q''_{x,y}$$

This scalar $BQM_{final}$ then contributes to the optimization objective that seeks to maximize the integrated quality metric across all pattern cases and channels.

The innovative $L_{BQM}$ component of our model's loss function derives a metric from both the predicted and actual feature matrices. This component assesses the accuracy of transformations along with classifications and regressions through the aforementioned affine transformations and bilinear interpolations:

$$L_{BQM} = \text{MSE}(BQM_{pred}, BQM_{label})$$

Finally, the comprehensive loss function integrates this component:

$$L = \alpha \cdot L_{\text{regression}} + \beta \cdot L_{\text{classifier}} + \gamma \cdot L_{BQM} \tag{2}$$

## 5 CNN Modeling Results

We used PyTorch to train the network in Fig. 5. Our dataset consisted of synthetically generated pulse responses as depicted in Fig. 3(a). We generated 1024 unique channels with $h_1, h_2, h_3, h_4$

coefficients sampled from uniform distributions. Furthermore, we generated 32 more minor variations with different transmit patterns for a total of 32,768 data points. Each data point consists of a 3D tensor capturing the BQM across 16 ($m = 4$) pattern cases and the ILP solutions. ILP solutions were carried out both $k = 2$ and $k = 4$ level cases. The training and validation datasets were drawn from Channels 1-950, with the remaining 74 being reserved for the test dataset. PyTorch jobs were run on NVIDIA V100 GPUs in a DGX-1 configuration.

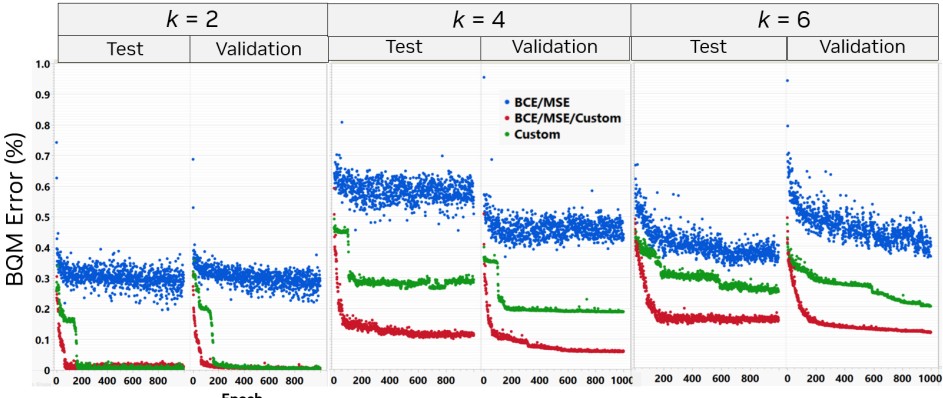

Figure 6: CNN BQM Performance Metrics for $k = 2, 4, 6$

Figure 6 shows the BQM performance for $k = 2, 4, 6$. Ablation studies were conducted to optimize the weighting parameters $(\alpha, \beta, \gamma)$ in Eqn. 2. The results in Fig. 6 demonstrate that the conventional loss metrics (BCE+MSE) perform significantly worse compared to our custom BQM loss metric across all $k$-levels. For $k = 2$, both the custom-only loss ($\alpha = \beta = 0$) and BCE/MSE/Custom combinations achieve comparable performance. However, for $k = 4, 6$, the BCE+MSE terms are critical for finding better solutions, as the custom-only loss struggles to match performance in these harder cases.

| BQM % | $k = 2$ | | | | $k = 4$ | | | | $k = 6$ | | | |
|---|---|---|---|---|---|---|---|---|---|---|---|---|
| | $\mu$ | $\sigma$ | CI-Low | CI-Up | $\mu$ | $\sigma$ | CI-Low | CI-Up | $\mu$ | $\sigma$ | CI-Low | CI-Up |
| BCE/MSE | 29.08 | 2.81 | 28.92 | 29.24 | 58.27 | 2.83 | 57.9 | 58.7 | 37.30 | 2.16 | 37.00 | 37.60 |
| BCE/MSE/Custom | 1.35 | 0.37 | 1.35 | 1.37 | 11.39 | 0.38 | 11.33 | 11.44 | 15.97 | 0.61 | 15.89 | 16.06 |
| Custom | 0.31 | 0.4 | 0.29 | 0.34 | 28.83 | 0.53 | 28.76 | 28.91 | 25.34 | 0.46 | 25.28 | 25.41 |

Table 3: BQM % Error Metric Statistics Across $k = 2, 4, 6$ Level Cases

Table 3 presents a statistical analysis of the BQM error percentages across $k = 2, 4, 6$-levels, with 95% confidence intervals. While the performance degrades for $k = 4, 6$, this is partly due to dataset complexity and ILP timeouts, which impact label quality, it still meets our application criteria. While larger networks could potentially improve results, hardware constraints discussed in Section 6 limit such options.

# 6 HARDWARE IMPLEMENTATION USING MICROCONTROLLERS

With our successful CNN training approach, the next step is to integrate it into hardware. Ideally this will reside in the interface controller hardware which usually consists of a microprocessor. Over the last few years, there has been a concerted effort to bring some of the ML based hardware acceleration techniques to microprocessors. For example, the Risc-V specification recently added a vector extension to han-

| Metric | ILP Solver | CNN (This Work) |
|---|---|---|
| Platform | High-end CPU | RISC-V micro-controller |
| Execution Time | ~120 s | ~5 ms |
| Memory | GBs of DRAM | ~1 MB SRAM |
| Update Rate in HW | Impractical | 1 second (for multiple lanes in micro-controller) |

Table 4: Comparison of ILP Solver vs. CNN Hardware Requirements

dle some deep learning calculations Kovačević
et al. (2022).

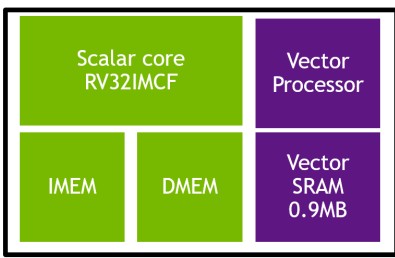

Figure 7: $\mu$controller Architecture: Risc-V core, instruction and data memory (IMEM/DMEM), and vector extension blocks (purple). Vector SRAM (0.9MB) is allocated for CNN weights (716 KB), buffers (148 KB), and kernels (30 KB)

| Operation | Input Dim | Output Dim | Cycles |
|---|---|---|---|
| Conv1 + ReLU | 16 x 32 | 32 x 64 | 722,944 |
| Skip1 | 64 x 16 | 128 x 16 | 165,888 |
| Conv2 + ReLU | 64 x 16 | 128 x 64 | 1,476,608 |
| Skip2 | 128 x 8 | 256 x 8 | 165,888 |
| Conv3 + ReLU | 128 x 8 | 256 x 8 | 1,476,608 |
| Max Pool/Add | - | - | 2,048 |
| Global Average Pool | - | - | 256 |
| **Total Cycles** | - | - | **4,027,680** |
| **Effective Cycles w/ Margin** | - | - | **5,034,600** |
| **Total Cycle Time @ 1GHz** | - | - | **5.08ms** |

Figure 8: Cycle count for CNN operations on Risc-V $\mu$Controller with Vector Extension

To assess hardware feasibility, we estimated the hardware requirements based on the network shown in Fig. 5. Figure 7 shows our floorplan for the microcontroller design using a Risc-V architecture. The purple boxes are the blocks added to support vector processing to enable more efficient inference computation. Figure 8 provides the estimated cycle count assuming Risc-V vector extension support for a single pass through the network. Assuming a modest clock frequency of 1GHz, the total cycle time would be $\frac{1}{N_{cycles}*T_{clk}} = 5.03mS$. Given that our goal is periodic updates to compensate for slow temperature drifts, this cycle time is more than adequate for our periodic training given

While this demonstrates initial feasibility, obtaining accurate power and latency numbers would require substantial cross-functional design. However, our initial analysis shows that the added vector processor and SRAM for CNN inference introduce an estimated power overhead of approximately $10mW$ per microcontroller. This overhead is outweighed by two key energy-saving mechanisms:

- **Sparse ADC Design**: By reducing the number of non-uniform slice levels, our approach significantly lowers ADC power consumption compared to dense, uniform designs. For example, in flash-based ADCs, power scales proportionally with $k$, offering substantial savings for lower $k$ configurations.

- **Per-Lane Power Optimization**: Many lanes in high-density links exhibit higher performance across tests, allowing us to configure some lanes with as few as one slice level ($k = 1$). As shown in Fig. 9(c), this approach preserves eye area margins while optimizing power on a per-lane basis, akin to the waterfilling technique in wireless communications (Cioffi, 2023). These tailored configurations enable significant energy efficiency improvements in terms of I/O per mm and pJ/bit.

These techniques collectively reduce receiver power consumption, making the CNN overhead justifiable and reinforcing the practicality of our proposed approach for high-density link applications.

## 7 APPLICATIONS AND POTENTIAL BENEFITS

Looking ahead, we showcase some of the potential benefits for AI computing. To illustrate the high-speed link density improvements, we collected measurement data on a GDDR memory interface (18Gbps) as shown in Fig 9(a). This system was an early version of a Notebook platform where there was high crosstalk on a few signals.

We utilized our scheme on the weakest bits impacted by large crosstalk with $k = 2$ slice levels. Fig. 9(b) shows the BQM gain when using crosstalk terms. As shown in Fig. 9(c), lanes with higher performance can be configured with minimal slice levels ($k = 1$), effectively reducing power consumption while maintaining error-free operation.

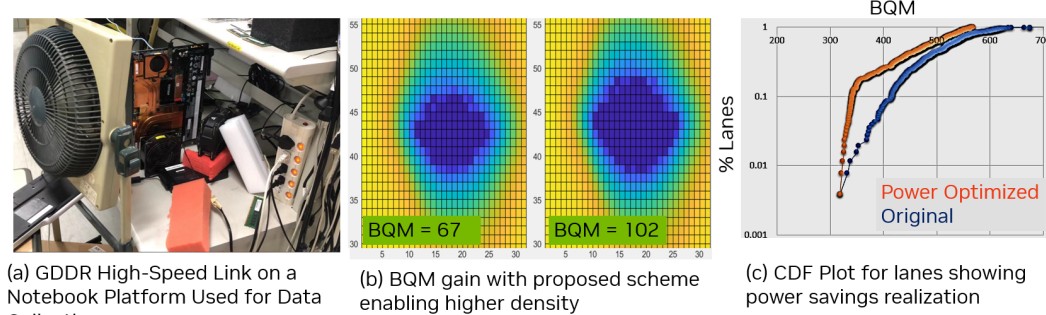

(a) GDDR High-Speed Link on a Notebook Platform Used for Data Collection

(b) BQM gain with proposed scheme enabling higher density

(c) CDF Plot for lanes showing power savings realization

Figure 9: Prototyping our proposal using lab data (a) Notebook memory setup (b) BQM gain with crosstalk consideration (c) Power savings optimization by reducing $k$ for stronger lanes

These experiments also highlight the robustness of our approach: a **model trained on synthetic data** can be **successfully applied to real-world silicon links**. This ability is crucial for practical applications, as it allows us to leverage the ease of synthetic data generation while still achieving performance gains on real hardware.

## 8    KNOWN LIMITATIONS AND FUTURE WORK

While this paper lays the groundwork for more intelligent high-speed links, significant cross-functional development is needed to realize the concept in silicon. We outline several key considerations:

- **Low-Resolution and Low-Power ADC Design**: Designing an effective low-power ADC is complex and could constitute a separate research project. This paper demonstrates up to $k = 6$ levels, corresponding to an effective number of bits (ENOB) of 2.56-bits, but such a design will require tradeoffs.

- **Current Focus on Electrical Links**: This study primarily validates the proposed framework on shorter-reach electrical links, which typically exhibit minimal non-linearities. However, the approach is adaptable to systems with significant non-linear behaviors, such as optical communication channels (Petermann, 2015), provided that the input matrix data accurately reflects these dynamics. The flexibility of the CNN and LUT structure ensures that the framework can generalize to such non-linear systems with appropriate characterization and input matrix generation (please see Appendix A.4).

- **Noise Considerations**: Although random noise was injected into the synthetic datasets, high-speed links can experience various uncorrelated noise sources. These noise profiles may differ between training sequences and live operation, necessitating margining to accommodate potential variations.

## 9    CONCLUSION

In this work, we proposed a multi-task CNN framework for optimizing high-speed link performance, addressing challenges in part-to-part variations and time-varying effects like temperature drift. By leveraging a custom loss function and integrating CNN inference into microcontrollers, our approach achieved high accuracy and substantial reductions in link error metrics, demonstrating practicality for high-speed link applications. Looking ahead, we aim to extend this framework through prototyping, analog design innovations, and tighter integration of AI-driven optimization in I/O controllers, advancing energy-efficient, adaptive high-speed link architectures.

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

# A APPENDIX

## A.1 RECEIVER DESIGN AND TARGET LINKS

Figure 10 illustrates representative receiver architectures for short-reach links (a) and long-reach links (b). In Fig. 10(a), the design employs a simple 1-tap decision feedback equalizer (DFE) with basic analog-to-digital conversion using low complexity "samplers".

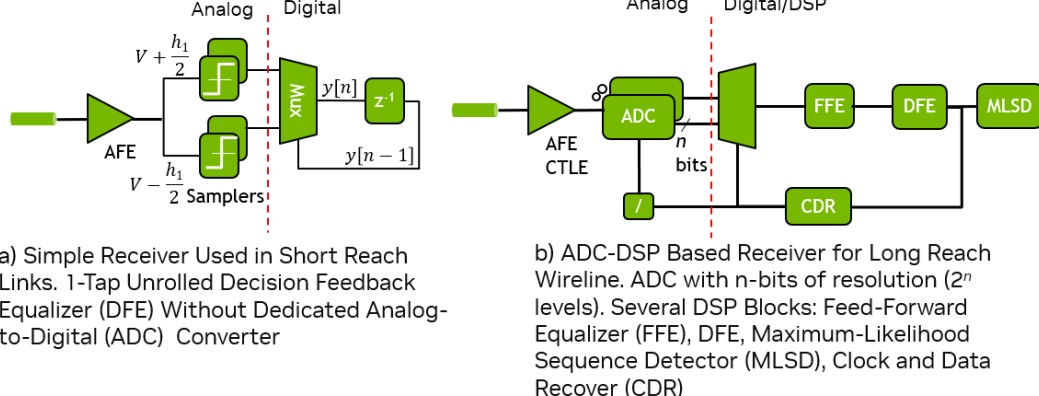

a) Simple Receiver Used in Short Reach Links. 1-Tap Unrolled Decision Feedback Equalizer (DFE) Without Dedicated Analog-to-Digital (ADC) Converter

b) ADC-DSP Based Receiver for Long Reach Wireline. ADC with n-bits of resolution ($2^n$ levels). Several DSP Blocks: Feed-Forward Equalizer (FFE), DFE, Maximum-Likelihood Sequence Detector (MLSD), Clock and Data Recover (CDR)

Figure 10: Illustration of Benefit When Increasing Observed Feedback Terms on Voltage Margin (a) Channel pulse response (b) CDF without equalization (c) CDF using conventional 1-tap DFE (d) CDF with proposal (e) Conventional DFE schematic (f) Proposal schematic

In contrast, Fig. 10(b) shows a more complex receiver architecture used for longer-reach links like Ethernet. It incorporates time-interleaved analog-to-digital converters (ADCs), where each ADC has $n$ bits of precision, resulting in $k = 2^n$ levels. Typically, these ADC levels are uniformly spaced. The additional DSP stages—such as Continuous-Time Linear Equalizers (CTLE), Feed-Forward Equalizers (FFE), Decision Feedback Equalizers (DFE), and Maxium Likelihood Sequence Detectors (MLSD) are necessary to mitigate ISI over longer distances. This added complexity comes at the cost of increased power consumption and silicon area. Our focus in this work is on links with receivers similar to Fig. 10(a).

## A.2 THEORETICAL DISCUSSION ON ERROR BOUNDS

Our CNN is trained to approximate the solution of an ILP problem by learning from optimal ILP solutions generated offline. While we demonstrated that the CNN achieves good performance, a more formal proof of the error bounds between the ILP and CNN solutions has not yet been derived.

Under certain conditions, the optimization problem solved by the ILP guarantees optimality. However, the CNN-based solution, introduces approximation errors due to the following factors:

- **Neural Network Generalization Error:** The CNN learns a mapping from inputs to optimal ILP solutions, but the approximation may deviate from optimality, specifically for unseen test cases. We did shows results on the unseen test case, but we make the general observation here.

- **Universal Function Approximators:** While neural networks are universal approximators, the limited capacity of the network (depth and neuron count) and training data limitations may introduce errors.

Formally, let $S_{ILP}$ represent the optimal solution derived from the ILP, and let $S_{CNN}$ be the solution predicted by the CNN. We are interested in deriving a bound on the approximation error $\|S_{ILP} - S_{CNN}\|$ under some relevant metric factoring in the multi-task solution nature.

While a closed-form bound on $\|S_{ILP} - S_{CNN}\|$ has not been established, the approximation error can be tied to the CNN's ability to minimize the custom loss function during training. We hypothesize that error bounds could be formulated based on the following factors:

- **Network Capacity:** Larger CNN architectures may provide tighter approximations of the ILP solutions. Of course, in our practical implementation, we have the additional constraint of fitting within a microcontroller.

- **Training Data:** A more extensive dataset can improve the network's generalization, potentially reducing the error. We mentioned this in the paper for the $k = 4$ case.

- **Loss Function Behavior:** The custom loss function, which mimics the ILP objective, plays a big role in the approximation error (along with the more conventional losses). A formal analysis of the custom loss function including the affine translation and Gumbel-Softmax may help to bound this error.

## A.3 ILP PSEUDO-CODE

We detail our pseudo-code to find the optimal BQM with the input per-pattern matrices below in Algorithm 1.

---

**Algorithm 1** Optimization of Receiver Parameters via ILP

---

1: **Define Inputs:**
2: 3D Matrix $err\_mat$ which is $2^m \times n \times p$ consisting of error information for each pattern case
3: **Define Parameters:**
4: Number of pattern cases $2^m$, voltage steps $n$, phase steps $p$, unique levels $k$.
5: **Define Decision Variables:**
6: Binary $X[i, l]$ for each pattern $i$ and level $l$, indicating if level $l$ is chosen for pattern $i$.
7: Binary $W[j, z]$ for each voltage-time coordinate $(j, z)$, indicating if coordinate is error-free.
8: Binary $U[l]$ for each level $l$, indicating if level $l$ is active
9: **Objective:**
10: Maximize error-free coordinates: Maximize $\sum_{j=1}^{n} \sum_{z=1}^{p} W[j, z]$.
11: **Constraints:**
12: Ensure exactly one level per pattern: $\forall i, \sum_{l=1}^{n} X[i, l] = 1$.
13: Link patterns to levels: $\forall i, l, X[i, l] \leq U[l]$.
14: Maintain $k$ total active levels: $\sum_{l=1}^{n} U[l] = k$.

---

## A.4 LINEARITY DISCUSSION

Our ILP formulation itself does not assume linear quantization levels, as demonstrated in the pseudocode in Algorithm 1 of the paper. Instead, it optimizes based on the input BQM. If the BQM is generated using linear ADC offset assumptions, this may influence the labels provided by the ILP

solver. However, the framework is adaptable to non-linearities, provided the BQM accurately captures them. The CNN, trained on the BQM data, is designed to learn patterns and variations inherent in the input data. While the current study validates the framework on electrical links with minimal non-linearities, the approach is flexible and can be extended to handle more pronounced non-linear behaviors.

### A.4.1 FRAMEWORK'S FLEXIBILITY

The LUTs in the receiver allow for non-linear equalization by enabling independent adjustment of slice levels for each pattern. For instance, a 0-1-0 ($m = 3$) pattern need not correspond to the exact "negative" slice level of a 1-0-1 pattern. This flexibility supports a range of non-linear behaviors in the ADC, receiver, or transmitter. Including this discussion clarifies that the framework is not restricted to linear systems and is.

### A.4.2 FUTURE EXTENSIONS

While the current work focuses on electrical links, systems such as optical channels, which exhibit larger non-linearities, represent a complementary research direction. Discussing non-linearities provides a foundation for extending the framework to these systems in future work. Scope and Completeness of the Current Study:

### A.4.3 VALIDATION ON REAL-WORLD DATA:

The framework has been validated on real lab-measured BQM data from electrical links as discussed in Section 7 of the paper, where there likely are some non-linearities, albeit minimal. . Electrical links are ubiquitous in modern computing platforms (memory interfaces, chiplet based interfaces, GPU-GPU, CPU-GPU inter-chip links off-chip, Networking, ...) so solving this problem is impactful and practical on its own.

### A.4.4 JUSTIFICATION FOR SEPARATE VALIDATION OF NON-LINEARITIES

Testing the framework under significant non-linearities, such as those found in optical systems, would require substantial extensions to the current study, including:

- Generating per-pattern BQM data for optical channels with high non-linearities.
- Training and validating the CNN on datasets that reflect these unique system characteristics.

These extensions are large enough to warrant a dedicated paperas incorporating them into the current work would detract from our existing contributions.

### A.5 SIGNAL DETECTION

For digital communication, we need to sample this continuous time waveform to convert the data to bits. Figure 11(a) captures the resulting time-domain receiver response for a signal after launching a pulse on its own (ISI) and neighboring transmitter (crosstalk). Transitioning to a discrete time statistical model, the voltage probability density function (PDF) for the receiver at a given sampling time can be computed by factoring pattern probabilities along with the conditional channel probability $p_{y|x}$. To calculate the probability of error, we consider a simple signaling scheme where we only send a "0" or "1" and use a single threshold ($k = 1$). An error occurs when $y(t')$ crosses the threshold $v_{\text{ref}}$ incorrectly relative to the binary value of the main input signal $P(\text{error} \mid x_{\text{main}}(t') = 0) = P(y(t') > v_{\text{ref}} \mid x_{\text{main}}(t') = 0)$ and $P(\text{error} \mid x_{\text{main}}(t') = 1) = P(y(t') < v_{\text{ref}} \mid x_{\text{main}}(t') = 1)$. The optimum signal detector chooses the message which minimizes the probability of error, which can be thought of as a maximum a posterior detector (MAP) Cioffi (2023). In the case where the pattern probabilities ($p_x$) are equal, this reduces to a maximum likelihood detector. Figure 11(b) shows the resulting voltage PDF and its integrated CDF. In this paper, we investigate the benefit of using additional threshold levels ($k > 1$) along with simple boolean functions on a signal and its neighbors' history to improve link performance.

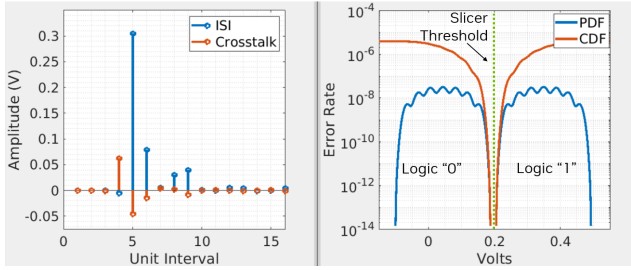

Figure 11: MIMO channel model: (a) Pulse response (b) Statistical evaluations

## A.6 Mathematical Framework & Receiver Proposals

The voltage at the receiver, $y(t)$, is a linear superposition of prior transmitted bits (called ISI), crosstalk from nearby lanes, and noise. Let $p_{j,\nu}(t)$ represent the received voltage on channel $\nu$ at time $t$ for a pulse transmitted at time 0 from channel $j$. For $j \neq \nu$, this is crosstalk; for $j = \nu$, it is the channel pulse response. The response from an example channel is shown in Figure 12.

To account for the influence of previous bits (up to $M$ symbols) and sum over all channels $j$ to capture crosstalk, we express the received voltage as:

$$y_\nu(nT+t) = \sum_{j=1}^{J} \sum_{m=0}^{M} x_j[n-m] \cdot p_{j,\nu}(t+mT) + \eta(t)$$

(3)

where $x_j[i]$ are the transmitted symbols on channel 'j', $J$ is the number of lanes, $M$ is the number of prior symbols, $T$ is the symbol period, and $\eta(t)$ is random noise.

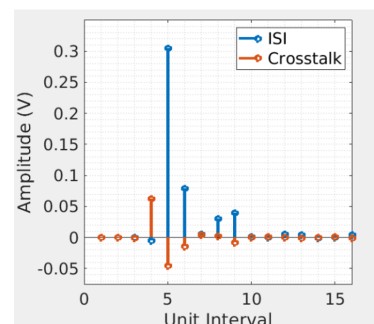

Figure 12: MIMO (multiple-input multiple-output) channel model pulse response

Looking at Eqn. 3, it is easy to see that preceding bits ($x_j[n-m]$) influence $y_\nu(nT)$ by shifting the eye by $p_{j,\nu}(mT)$. This means one can get better margins by moving the slice level depending on the prior bits, the basis of DFE Cioffi (2023) which is explained next.

## A.7 Two Slice levels

We first explore increasing the 1D voltage margin metric using $k = 2$ levels and previous decisions to determine which level to use. This metric can be visualized by taking a vertical slice of the eye diagram in Fig. 1(c), and use the x-axis to plot the error rate on a log scale. Consider a channel with a time-domain response shown in Fig. 13(a), where the main signal amplitude is $h_0$ with three dominant ISI cursors $h_1, h_2, h_3$. Using a single threshold (Fig. 13(b)) yields minimal voltage margin. A conventional unrolled decision feedback equalizer (DFE) typically uses $2^m$ levels for $m$ noise sources Stojanovic et al. (2005). With $k = 2$, only one noise source can be targeted, so we cancel $h_1$ by setting slice levels to $\pm\frac{h_1}{2}$ and using $y[n-1]$ to select the correct level (Fig. 13(e)). As seen in Fig. 13(c), the voltage margin improvement corresponds to $h_1$. To also mitigate $h_2$ and $h_3$, a conventional unrolled DFE would require $k = 2^m = 8$ levels. Since adding levels increases power and complexity, we ask: *Can we increase the margin with $k = 2$ by observing $y[n-1], y[n-2], y[n-3]$?*

This is feasible if the sum of any subset of non-dominant terms exceeds the dominant term:

$$j = \arg\max(|\mathbf{h}|), \quad \sum_{i \neq j} |h_i| > |h_j|$$

(4)

Given $h_2 + h_3 > h_1$, observing all three feedback terms should enhance the margin, as shown in Fig. 13(d). The LUT in Fig. 13(f) has $2^m = 8$ entries, simplifying the logic to a majority voting function among $y[n-1], y[n-2], y[n-3]$. While $k = 8$ levels allow precise voltage margin

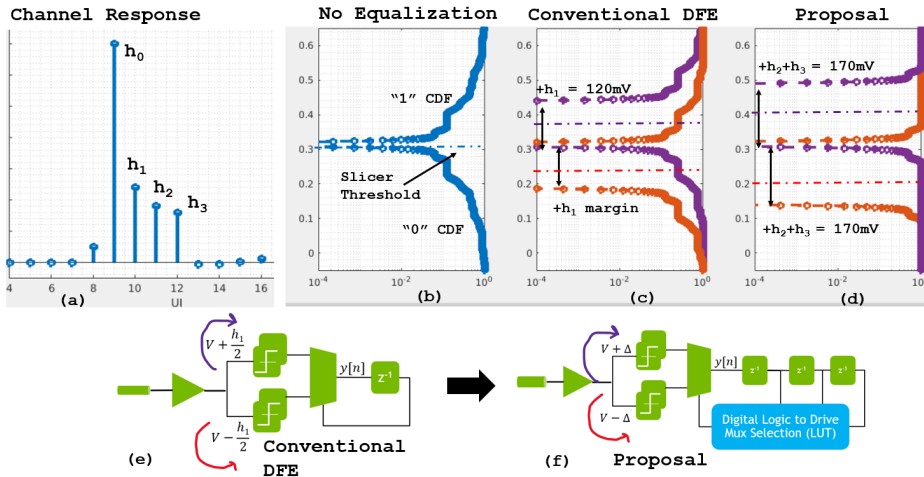

Figure 13: Illustration of Benefit When Increasing Observed Feedback Terms on Voltage Margin (a) Channel pulse response (b) CDF without equalization (c) CDF using conventional 1-tap DFE (d) CDF with proposal (e) Conventional DFE schematic (f) Proposal schematic

maximization, our goal is to enhance efficiency with minimal $k$. We demonstrate that with the same complexity of a $k = 2$ ADC, performance improves by considering additional observations. This approach generalizes to any $k$ levels and $m$ feedback taps, but we need to introduce a more suitable metric for link performance before formally defining our optimization problem.

## A.8  2D AREA MARGIN METRICS & ERROR COUNTERS

Given there are numerous sources of both voltage and timing error in links, we propose using the eye area, a **2D** area metric, since it indicates how much uncertainty we can tolerate in both dimensions. From an implementation perspective, we can measure this by doing a nested sweep across *(time, voltage)* and track the errors in a training sequence as shown in Fig. 14. We have error counters that correspond to the various pattern cases which we are observing ($y[n-1], ...y[n-m]$), resulting in $2^m$ counters. While the representation of margin in the continuous time domain as in Eqn. 3 is referred to as an eye diagram, we introduce the term *bivariate quality metric* (BQM) since it is after sampling. This metric is defined as $\sum_v \sum_t BER(v, t) < \kappa$ or the number of points in a 2D grid of voltage and timing points which meet a target bit error rate (BER) $\kappa$.

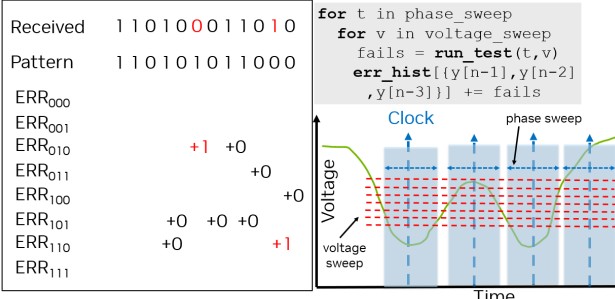

Figure 14: Bi-variate Quality Metric (BQM) implemented using a nested 2D Sweep with hardware error counters

## A.9  ON-CHIP SAMPLING SCOPE

For most link characterization efforts, there are error counters which indicate whether a specified bit sequence has errors. Usually sweeps are performed in the voltage and time domains to check the

margin in the eye. While this method is quite effective in characterizing link margin, one drawback is that there is no indication for which symbols were erroneous. To address this limitation, we developed a virtual on-chip sampling scope. Similar to the link characterization approaches, we sweep the eye in both voltage and time dimensions, but we now have a register to record the bit stream. The pseudo-code is listed in Algorithm 2.

---

**Algorithm 2** On-Chip Sampling Scope Pseudo-Code

---

1: **for** $i \leftarrow 1$ to $iterations$ **do**
2:     ProgramBurstLocation() {to send/receive}
3:     InitializeLink()
4:     TrainLink()
5:     **for** $x \leftarrow 1$ to $timingSteps$ **do**
6:         **for** $y \leftarrow 1$ to $voltageSteps$ **do**
7:             ReadPatternFromMemory()
8:             SaveToRegisters()
9:             PollRegisters()
10:         **end for**
11:     **end for**
12: **end for**

---

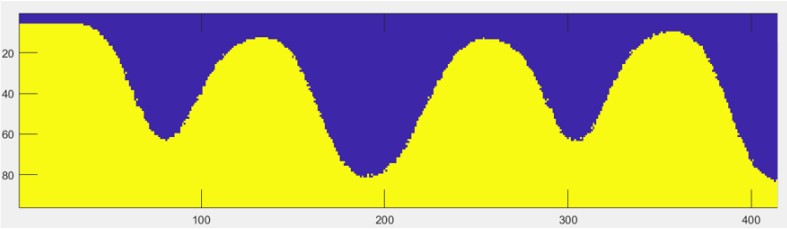

Figure 15: On-Chip Sampling Scope

