# OpenReview forum: "SERDES Link Training with Edge Inference: Neural-Network Driven Discrete Optimization to Maximize Link Efficiency"
_ICLR.cc/2025/Conference — Submitted to ICLR 2025_

### Official Review · Reviewer_cjzU · 2024-11-01

**Soundness:** 2
**Presentation:** 3
**Contribution:** 3
**Rating:** 6
**Confidence:** 4

**Summary:**

This paper introduces a machine learning framework for leveraging convolutional neural networks (CNNs) to dynamically optimize the parameters of high-speed SERDES link receivers, thereby mitigating signal attenuation and noise to enhance data communication performance. The framework employs Integer Linear Programming (ILP) to generate training labels for the CNN, and then constructs a multi-task CNN architecture. Also this paper develops a custom loss function that significantly enhances receiver adaptability and performance. Experimental results validate the efficacy of this approach in real hardware environments, demonstrating substantial improvements in energy efficiency and crosstalk mitigation. Future research will concentrate on advanced hardware implementation and the seamless integration of deep learning with I/O controllers.

**Strengths:**

This paper presents a significant innovation by integrating artificial neural networks into the parameter optimization of high-speed interface SERDES. The novel approach employs CNNs to replace the conventional ILP solving process, marking a substantial advancement in the field. Additionally, the article exhibits a clear and coherent writing structure, with well-articulated theoretical arguments. If the outcomes are indeed as stated, this work holds considerable importance and relevance for optimizing high-speed interface SERDES.

**Weaknesses:**

In the experimental demonstration section of this paper, the reliability of the findings is questionable. The authors assert that the ILP solver consumes significant hardware resources, rendering it impractical for high-speed SERDES interfaces; hence, they propose using a CNN instead. However, it is well-established that CNN architectures can vary considerably in terms of hardware resource requirements depending on their size and complexity. The paper does not specify the size of the CNN model employed, nor does it provide a comparative analysis of the hardware resource consumption between the CNN and the ILP solver. Consequently, the claim that CNN is more hardware-friendly lacks substantiation.

Furthermore, the paper contains an excessive amount of information regarding the ILP method itself, which detracts from the primary focus of the study. Instead, the authors should emphasize the specific advantages of utilizing a CNN, supported by experimental data rather than relying solely on theoretical derivations. By providing more convincing evidence and clearer explanations, the paper could enhance its overall credibility and impact in the field.

**Questions:**

Please conduct an experiment to compare the hardware costs associated with the ILP solver and the CNN network. Additionally, it would be beneficial to redraw the CNN network architecture diagram, as the current use of direct screenshots from the ONNX model significantly affects the clarity of the presentation. A clearer and more precise illustration would enhance the understanding of the network's structure.

---

### Official Review · Reviewer_ohG3 · 2024-11-04

**Soundness:** 3
**Presentation:** 2
**Contribution:** 2
**Rating:** 3
**Confidence:** 4

**Summary:**

This paper presents an edge inference framework designed to optimize non-uniform quantization levels in programmable analog-to-digital converter (ADC) receivers for high-speed communication links. Instead of using an ILP-based solver, the authors train a convolutional neural network on ILP-derived solutions, allowing for efficient edge inference. The CNN works decently on synthetic data, and is evaluated on a real system with a simple task, k=2.

**Strengths:**

Strength:
* The trained model is evaluated on a real system and shows its improvement over the un-optimized version.

**Weaknesses:**

Weakness:
* The paper lacks ML novelty and may not be suitable for ICLR. For example, in Section 4.1, the Gumbel-Softmax trick is a widely used trick for discrete variables. The CNN model also has no fundamental contributions. This paper is more suited for the circuit community.
* The experiments part is incomplete, and more results are required. In section 6, the author only evaluates the model on a simple case (k=2); a higher k, like 4, is not presented in this paper. Moreover, the author should report the golden baseline using ILP as a reference to assess the methods' effectiveness.
* The target devices have only up to 2-bit ENOB, questioning the method's effectiveness for practical ADC with higher ENOB. Based on the results, the proposed model shows a much higher test loss on k=4 compared to k=2. Based on this observation, the method should perform much worse on higher ENOB cases.

**Questions:**

Questions
* The ML contribution of this paper is not significant and needs more justification.
* The demonstrated ADC has a too-small ENOB. Evaluating on more challenging, high-ENOB ADC is desired.

---

### Official Review · Reviewer_oczn · 2024-11-04

**Soundness:** 2
**Presentation:** 2
**Contribution:** 2
**Rating:** 5
**Confidence:** 3

**Summary:**

The growing data size of AI applications requires the data communication link to have high throughput and efficiency. Desirably, each lane of high-speed communication link should have its own bit pattern for optimized quantization level, but the high computational cost of ILP makes it unsuitable for on-chip deployment. The paper proposed  an edge inference framework that leverages integer linear programming (ILP) and NN-based discrete optimization to optimize non-uniform quantization levels in programmable analog-to-digital converter (ADC) receivers.

**Strengths:**

The paper proposed  an edge inference framework that leverages integer linear programming (ILP) and NN-based discrete optimization to optimize non-uniform quantization levels in programmable analog-to-digital converter (ADC) receivers.
To address the computational cost of ILP solvers, they train a CNN to mimic the ILP solver. Their custom loss function, which outperforms standard metrics like cross-entropy and mean squared error (MSE), reduces area metric errors from 29% to less than 2%

**Weaknesses:**

The idea seems to be similar to knowledge distillation. Needs more justification for the novelty.

**Questions:**

Why does the optimization of quantization levels need to be put on-chip rather than being an offline optimization?
They mentioned they train the CNN with the ILP solver’s input/output being the training data. How did the input of ILP get generated? How to make sure the training data is sufficient and divergent?
How to verify that the trained CNN can outperform compared with the original ILP solver rather than just being overfitting for the created dataset?

---

### Official Review · Reviewer_8Rcv · 2024-11-04

**Soundness:** 3
**Presentation:** 3
**Contribution:** 3
**Rating:** 5
**Confidence:** 2

**Summary:**

This paper presents a novel approach for optimizing high-speed data communication links using neural networks for edge inference with the focus on optimizing quantization levels in Analog-to-Digital Converters (ADCs) in SERDES (Serializer/Deserializer) receivers. This work introduces an Integer Linear Programming (ILP) framework to achieve high-quality optimization for quantization levels, which is then approximated by a Convolutional Neural Network (CNN) to enable efficient on-chip inference. By deploying this CNN on a RISC-V microcontroller, the system dynamically adapts to changing channel conditions, improving error rates and energy efficiency.

**Strengths:**

Novelty: The paper effectively combines ILP with a CNN to approximate the solution for a challenging discrete optimization problem, reducing computational overhead and enabling real-time adaptation.

Impact: Addressing the optimization of high-speed communication links has significant implications for improving AI model deployment and efficiency, making this work valuable for practical, large-scale applications.

Evaluation: Detailed evaluation on performance metrics like BQM, error rate, and classifier accuracy provides a comprehensive understanding of the proposed model's efficacy.

**Weaknesses:**

Potential Overhead: While the approach reduces computational cost, the implementation of CNNs on microcontrollers may still cause computational overhead, especially in extreme edge scenarios.

Assumptions in ADC Linearization: The optimization assumes strong linearity in ADC level offsets, which may not always be feasible in high-speed ADCs, potentially impacting the overall optimization performance in non-ideal conditions.

**Questions:**

Here are my two questions about this work:

- What are the computational and memory costs associated with implementing CNNs on edge hardware in this system? Specifically, how does this affect overall latency, energy efficiency, and potential throughput in high-speed communication environments?

- The approach assumes linearity in ADC quantization levels for optimal link performance. How sensitive is the system’s performance to deviations from this linearity? If the ADC exhibits non-linear behavior, what degradation in accuracy or efficiency might occur, and are there alternative methods to account for such non-linearity?

---

### Meta-Review · Area_Chair_6Vkb · 2024-12-13

**Metareview:**

The paper presents a framework combining integer linear programming (ILP) and convolutional neural networks (CNNs) for discrete optimization in high-speed SERDES links. However, the contribution primarily lies in integrating established techniques rather than developing new methods or insights. Techniques like Gumbel-Softmax and custom loss functions, while well-executed here, are standard and do not significantly advance the field of machine learning. The authors argue for novelty in combining these methods, but this integration lacks compelling theoretical or empirical contributions that would merit publication at a premier ML venue. Additionally, the experimental validation falls short of fully supporting the claimed contributions.

**Additional Comments On Reviewer Discussion:**

During the rebuttal period, reviewers raised concerns about the limited novelty of the ML techniques, insufficient experimental validation for higher k-level quantization, lack of clarity in hardware resource comparisons, and excessive focus on ILP methodology. The authors responded by emphasizing the integration of ML techniques for hardware-specific discrete optimization, adding results for higher k-values, and detailing hardware comparisons showing CNN’s efficiency over ILP solvers. They also streamlined ILP content and improved the CNN architecture diagram. Despite these efforts, the novelty remained incremental, and experimental evidence was still limited.

---

### Decision · Program_Chairs · 2025-01-22

Reject